# Effect of the Citrus Flavone Nobiletin on Circadian Rhythms and Metabolic Syndrome

**DOI:** 10.3390/molecules27227727

**Published:** 2022-11-10

**Authors:** Gael N. N. Neba Ambe, Carlo Breda, Avninder Singh Bhambra, Randolph R. J. Arroo

**Affiliations:** 1Leicester School of Pharmacy, De Montfort University, The Gateway, Leicester LE1 9BH, UK; 2School of Allied Health Sciences, De Montfort University, The Gateway, Leicester LE1 9BH, UK

**Keywords:** polymethoxy flavone, ageing, non-communicable diseases, chemoprevention

## Abstract

The importance of the circadian clock in maintaining human health is now widely acknowledged. Dysregulated and dampened clocks may be a common cause of age-related diseases and metabolic syndrome Thus, circadian clocks should be considered as therapeutic targets to mitigate disease symptoms. This review highlights a number of dietary compounds that positively affect the maintenance of the circadian clock. Notably the polymethoxyflavone nobiletin has shown some encouraging results in pre-clinical experiments. Although many more experiments are needed to fully elucidate its exact mechanism of action, it is a promising candidate with potential as a chronotherapeutic agent.

## 1. Introduction—Circadian Clock

The 24 h rotation of the earth along its axis exposes all terrestrial organisms to light and dark cycles as well as to daily temperature fluctuations. In this environment, having a mechanism that can anticipate these changes instead of merely responding, provides an evolutionary advantage. Hence, an endogenous ≈ 24 h circadian clock (from the Latin *circa diem* meaning ‘about a day’) allows an organism to coordinate physiological activities according to cycling changes in the environment, food availability and predator risk [1]. Although initially observed in plants by Jean Jacques d’Ortous de Mairan (1729), the first genetic proof of circadian clock existence was made in 1971 by Konopka and Benzer who isolated the first arrhythmic *Drosophila melanogaster* mutants [2]. Fifteen years later, the period gene (*per*) was cloned in fruit flies [3] and another ten years later, the second circadian gene *timeless* (*tim*) was identified in fruit flies and soon thereafter also in mice [4,5]. Hall, Rosbash and Young, who were awarded the Nobel prize for physiology and medicine in 2017, elegantly demonstrated that the circadian rhythmicity is generated and sustained by transcriptional and translational feedback loops in which PER/TIM complexes inhibits their own CLOCK:CYCLE (CLK/CYC) activators [6,7,8]. A second interconnected feedback loop made by *vrille* and *Pdp1ε* (Par domain protein 1ε) sustains rhythmic transcription of *clk,* thus enhancing the stability of both cycles [9]. This model, initially described in fruit flies, was shown to be highly conserved across the kingdoms ranging from cyanobacteria and plants to insects and humans, although the function of some genes may have diverged between organisms [10]. In mammals, the CLOCK:BMAL1 heterodimer activates *Per* and Cryptochrome (*Cry*) transcription. Then, as in *Drosophila*, the PER:CRY complex translocates back to the nucleus to repress its transcription via CLOCK:BMAL1 interaction [11]. This primary negative feedback look is sustained by a second regulatory loop in which BMAL1 cyclic expression is maintained by the ROR (α, β and γ) activator and REV-ERB (α and β) repressor proteins [12] (Figure 1). Approximately 24 h are required to complete a full circadian cycle but several posttranscriptional and posttranslational events finely regulate these oscillators [13]. For example, casein kinase 1ε/δ (CK1ε/δ), plays a fundamental role in the establishment of a new circadian cycle by phosphorylating PER and CRY. Upon PER and CRY degradation by the 26 S proteasome complex, the suppression of CLOCK:BMAL1 activity is released allowing the cycle to start again [14]. Large gene expression profiling analyses have allowed the identification of several circadian-output genes [15,16]. While roughly 50% of genes in mammals show a level of circadian expression, researchers have highlighted that their profiles exhibit tissue-specific rhythms [17]. Strikingly, clocks have shown to regulate a multitude of pathways within the cells including their epigenetic profile, phosphorylation and metabolic profiles, and the microbiome in the organism [18,19,20,21,22].

In mammals, as in other multicellular organisms, virtually all tissues possess circadian oscillators, making the system organisation highly complex. The light-entrainable pacemaker is in the suprachiasmatic nucleus (SCN) of the hypothalamus and its function is to synchronize peripherical clocks. While all these oscillate within a period close to 24 h, it is essential that they are synchronized with the external environmental conditions. Hence, the key function of the SCN clock is to receive environmental light information by retinohypothalamic track and synchronize other molecular oscillators, both within the SCN and in peripheral organs [23]. While the synchronization signal is transmitted by neurotransmitters and neuropeptides within the brain regions, hormonal secretion and neural innervation are used to synchronize the peripheral tissues [24]. Melatonin and glucocorticoids (Figure 2) are two examples of the manifestation of the circadian clock in mammals. Light information received from the SCN is transmitted to the pineal gland for nocturnal melatonin secretion. Circulating melatonin can entrain peripherical clocks interacting with molecular clock mechanisms acting as a signal for the dark phase of the photoperiod [25]. In the adrenal glands, the secretion of the hormone glucocorticoid—regulating glucose homeostasis—is also under circadian regulation. Indeed, cortisol concentration levels peak during the morning and during active periods in diurnal organisms, while its concentration is reduced during the sleeping phase [26]. While photic cues are the main circadian synchronisers for the SCN pacemaker, other non-photic zeitgebers such as arousal stimuli (e.g., social interaction or exercise), food/feeding regimes and temperature can also act as cues in the peripheral clocks. Moreover, in this complex network, a hormone and metabolic signal-based bi-directional communication exists between the SCN and non-SCN oscillators, providing plasticity to the system and optimal adaptation to the environment [27,28].

Due to the impact of circadian oscillators on the physiology and behaviour of organisms, their dysregulations and disruptions are associated with the development of diverse pathologies. In mouse models, mutations or deletions in core circadian genes (e.g., BMAL1) cause increased levels of glucose and lipids, leading to premature ageing [29]. Interestingly, BMAL1 downregulation leads to tissue-specific dysfunctions causing metabolic and triglyceride biosynthesis impairments in the muscle, or obesity when knocked-down in adipose tissue [30,31]. Similarly, surgical removal of the SCN is linked to the onset of tumour growth and alteration in the microbiota and immune cells, in the intestine of mouse models [32,33]. In humans, one of the major contributions to circadian disorders is the misalignment between the endogenous clock and the environmental rhythms (such as the day-night cycle). Artificial light, shift work, travel and social lags are all clock misalignments introduced by the modern lifestyle. For example, night shift workers show an increased risk to develop several types of cancer, cardiovascular and metabolic disorders, psychiatric disorders, obesity and type-2 diabetes [34,35,36]. Traveling across multiple time zones, causes circadian rhythm desynchronization which ultimately leads to changes in sleep architecture, mood, hormone profiles, and gastrointestinal dysfunctions [37]. A great proportion of the teenage population experience social jet lag in which a variation in sleep pattern is observed between school/work days versus school/work free days. Notably, social jet lag is associated with an elevated consumption of alcohol and tobacco, as well as a higher incidence of obesity, diabetes, cancer, and cardiovascular disease [38,39]. Because circadian clocks regulate several cellular mechanisms such as oxidative stress, inflammation, neurotransmitter biosynthesis and metabolism, they have been linked to the development and progression of human neurodegenerative disorders [40]. While at present, the bi-directional relation between circadian disruption and neurodegeneration is not fully understood, evidence indicates that desynchronization of the clock over a lifetime, enhances the deposition of misfolded protein aggregates in Alzheimer’s and Parkinson’s diseases [41]. Moreover, in humans, single-nucleotide polymorphisms in core clock genes (e.g., CLOCK, BMAL1 and PER1) have been shown to increase the risk of Alzheimer’s and Parkinson’s Disease development [42].

Considering the importance of these oscillators, they are becoming novel therapeutic targets for the prevention or treatment of several pathologies. While these clocks can be enhanced and stabilised by non-invasive strategies such as phototherapy, restricted diet and exercise [43], synthetic and natural compounds are future candidates in circadian medicine, to help improve clock-regulated processes and treat clock misalignment diseases.

## 2. Effects of Natural Products on Circadian Rhythm

Phytochemicals, compounds derived from plants, have been known to influence a wide range of pharmacological processes. Many phytochemical compounds, such as flavonoids, alkaloids, polyphenols and melatonin, have been reported to have a regulatory effect on expression of genes linked to the circadian clock, and are thus expected to play a role in regulating the internal environment [44]. Flavonoids (Figure 3) in particular raised interest as compounds that may affect circadian rhythm and diseases related to appropriate regulation of circadian rhythm [45]. In spite of a wealth of information showcasing the role of flavonoids in various disease states, very little work has been done so far to specifically ascertain the effects of these molecules on the circadian physiology. Nevertheless, these studies are a rich source of information, and help us understand the significant role played by flavonoids in modulating different circadian systems. A flavonoid-rich fraction of the plant *Cyclocarya paliurus* was shown to have modulatory effects on both the liver clock genes as well as intestinal microbiota in a circadian rhythm disorder mice model. A robust rhythmic expression in most liver clock genes was observed, mainly *Clock1, Per1, Per2, Per3, Bmal1, Sirt1, Cry1* and *Cry2* over a 24 h period. Analysis of the plant extract showed kaempferol-3-*O*-β-glucuronide as the predominant flavonoid, with kaempferol-3-*O*-α-l-rhamnopyranoside, quercetin and quercetin-3-*O*-glucoside (isoquercitrin) found in varying proportions [46]. Other studies investigating the effect of flavonoids on circadian rhythms in different mice models showed similar outcomes [47,48,49,50]. Most flavonoids have poor bioavailability in mammals, and merely pass through the digestive tract to be metabolised by intestinal microbes in the colon. Therefore, some authors leave open the option that flavonoids may not directly affect mammalian physiology but do so indirectly by altering the gut microbiota which then, in turn, produce metabolites that affect the mammalian circadian system [46,50]. The concept of the gut-brain axis is intriguing and attracting increasing interest, though much still remains to be further explored.

Interestingly, mutations that alter *Arabidopsis* flavonoid metabolism, concomitantly affect the plant’s circadian clock [51]. This indicates that flavonoids can affect circadian rhythms in both the animal and plant kingdom, which may be seen as further confirmation that the mechanism of the circadian clock is highly conserved. Some flavonoids and their effects on the circadian variations are expanded on below:

The bioflavonoid quercetin has a significant effect on the sleep-wake cycle in male Sprague Dawley rats [52]. When sleep-wake states were classified as wakefulness, rapid eye movement (REM) sleep and non-REM sleep, a post hoc analysis of the rat’s diurnal cycle after quercetin administration (200 mg/kg) showed a notable decrease in REM sleep during the first three hours, an effect which was seen almost immediately. Additionally, a decrease in wakefulness and increased non-REM sleep during the last four hours were observed. The proposed pathway through which quercetin exerts its effects was the GABAergic pathway and GABA-independent mechanisms. Similarly, baicalein, a flavonoid found in *Scutellaria baicalensis*, affects the same pathway. It was shown to promote increase in sleep time via GABAergic non-benzodiazepine sites in mouse brains [53,54,55].

Quercetin further showed anti-metastatic activity against light/dark shift-induced metastasis in BJMC3879Luc2 mouse breast cancer cells transplanted into BALB/c mice [56], and was proven to affect circadian clock and age-related genes in fibroblast cells [57].and The flavonol is thought to have anti-obesity activity through the adenosine monophosphate-activated protein kinase (AMPK) signalling pathway [58], a pathway also known to be influenced by variations in the circadian clock [59]. Ditto, in mice (−)-epigallocatechin-3-gallate (EGCG), the major catechin found in green tea, appeared to regulate cell signalling pathways in the central nervous system (CNS) that are linked with diurnal rhythms. However, there was a lack in the diurnal rhythmic expression of the core clock genes *Bmal1* and *Clock* in the hypothalamus [60]. Similar studies reported AMPK-mediated ameliorative effects of EGCG in liver hepatocytes (HepG2 cells) [61].

Furthermore, the regulatory effect of EGCG on the circadian clock metabolic disorders resulting from a high-fat/high-fructose diet was evaluated, and it was observed that EGCG administration improved the diet-dependent decline in circadian function via the Sirt1-PGC1αloop [48,62].

Dietary proanthocyanidins (Figure 4) are the most abundant flavonoids in the Western diet. They have been found to have numerous beneficial effects on different diseased-states [63], and have also been proven to have a regulatory effect on glucose and lipid metabolism within the liver by adjusting the circadian rhythm via the modulation of *Nampt* gene expression, *Bmal1* acetylation and influencing NAD levels. Acute administration of a grape seed proanthocyanidin extract (GSPE) to male Wistar rats resulted in an increased expression pattern of the core clock genes, whereas a decrease in Nampt protein and mRNA levels was observed 3 h after administration [64]. In addition, GSPE treatments altered the oscillations of some plasma metabolites while no change in nocturnal melatonin levels was observed at ZT3 [65,66].

Discussed above are the most commonly occurring flavonoids which have been researched for their effect on the circadian cycle. However, the effect of other less-abundant flavonoids cannot be ignored. One of such is luteolin, a compound known to have neuroprotective ability [67] in diseases such as Alzheimer’s and epilepsy. This flavone also has the ability to promote sleep in male C57BL/6 mice [68]. In particular, oral administration of 3 mg/kg of luteolin, increased sleep durations up to 64.8 ± 1.2 min. An increase in NREM sleep by about 16.7% and decreased wake time by 40.9% was also reported.

Silybin A from *Silybum marianum*, was also shown to modulate the circadian clock by disrupting the CRY1-CLOCK interaction [69]. Additionally, Myricetin, known for its more hydroxylated structure and increased biological activity [70,71], significantly decreased serum melatonin levels and locomotor activity in nocturnal rats, by inhibiting the enzyme serotonin n-acetyltransferase [72]. In addition, Isorhamnetin has also been shown to affect circadian rhythms of DNA synthesis in the human oesophageal Eca-109 cell line [73].

## 3. Effects of Nobiletin Circadian Rhythm and Metabolism

Nobiletin (Figure 5) is a polymethoxy flavone that almost uniquely accumulates in the peel of *Citrus* fruits. Dried citrus peel, known as *Citri Reticulatae Pericarpium* or *Chenpi* in Chinese has traditionally been used to promote the circulation of qi (energy) throughout the body. The concept of ‘qi’ does not translate easily into western concepts of pharmacology, but *Chenpi* is used to sooth emotions including anger and irritability, and to supplement treatment of indigestion, and abdominal fullness through promotion of gastrointestinal motility [74].

A considerable body of work, done at the Department of Biochemistry and Molecular Biology of the University of Texas Health Science Center at Houston, has indicated that nobiletin can have a beneficial effect on human metabolism and acts through modulation of circadian clocks. Using a high-throughput chemical screening assay, based on fibroblasts expressing a *PER2::Luc* reporter gene construct [75,76], nobiletin was identified as a particularly effective clock amplitude-enhancing small molecule. Moreover, the assay shows that this flavone can directly affect the mammalian circadian system [44,77]. Furthermore, pharmacokinetic studies revealed significant brain and systemic exposure to nobiletin is feasible after oral administration [44]. We might speculate that this lipohilic flavone with its planar structure may mimick the activity of corticosterone (Figure 1), much like flavonoid phytoestrogens are known to mimick the activity of gonadocorticoids.

In high-fat diet-induced obese mice, treatment with nobiletin (200 mg/kg body weight via oral gavage every other day, for 10 weeks), significantly diminished body weight gain relative to the control group. In circadian clock-impaired Clock^Δ19/Δ19^ mutant mice that had been fed a high-fat diet, nobiletin treatment only resulted in a very modest reduced body weight. Throughout the circadian cycle, wild-type mice treated with nobiletin showed increased oxygen consumption compared with the controls, with the largest increase found in early dark phase; no such increase was observed in the Clock^Δ19/Δ19^ mutant mice [44].

Mice raised on a high-fat diet showed a build-up of intramuscular fat which in turn can impair energy homeostasis, partly as a result of a declined mitochondrial function. In addition, behavioural changes can be seen in the animals, e.g., reduced activity in circadian wheel-running assays [78]. If the mice were fed the same high-fat diet but with an added 0.1% nobiletin, fat accumulation in the muscle tissue was decreased and the animals were more active, showing circadian behaviour with clear increased activity at dawn and dusk [79]. Further mechanistic studies were done on skeletal muscle tissue isolated from mice, and on the mouse myoblast cell line C2C12. The data indicated that nobiletin activated mitochondrial OXPHOS gene expression [80] and in addition, nobiletin enhanced mRNA expression levels of mitochondrial glutathione peroxidase 1 (Gpx1) and thioredoxin 2, two enzymes that limit the build-up of reactive oxygen species (ROS), which are by-products of mitochondrial activity. Crucially, when *Ror-α*, *Ror-γ*, or *Ror-α/γ* genes were inactivated using CRISPR-guided DNA, cells failed to respond to nobiletin [80]. Further involvement of the circadian clock with mitochondrial activity in skeletal muscle was found following the metabolism of cardiolipin, a phospholipid that is critical for mitochondrial membrane structure and for integrity of mitochondrial respiratory chain complexes. Expression of the genes involved in cardiolipin biosynthesis shows circadian oscillation regulated by RORs; in aged muscle cells, as in *Ror*-deficient C2C12 cells, expression of *Taz* and *Ptpmt1* was disrupted or diminished. Nobiletin could partially restore transcription of these genes [81].

Skeletal muscle is the largest, mitochondria-rich metabolic organ and plays a key role in activity, thermogenesis, and overall energy homeostasis. However, other metabolic processes, like the urea cycle in the liver, are also controlled by the circadian clock. Nobiletin (200 mg/kg p.o. every other day) significantly lowered serum ammonia levels in mice that had been fed a high -protein diet. In the circadian clock-impaired mouse mutant, Clock^Δ19/Δ19^, the restorative effect of nobiletin was markedly reduced [82]. Further, in a murine Alzheimer’s disease model, real-time qPCR analysis revealed that nobiletin supplementation (0.1% in the normal diet) increased expression of several core clock genes in the mouse cerebral cortex, notably *Bmal1*, *Npas2* (a paralogue of *clock*), and *Ror-α*. Additionally, nobiletin activated various clock-controlled metabolic genes that are involved in insulin signalling and mitochondrial function. Thus, the flavone normalized exaggerated respiratory activity that is a symptom of Alzheimer’s. The normalizing effect was especially seen during the late dark phase in circadian cycle [83].

Using the *PER2::Luc* reporter gene expressed in transgenic mice, it could be shown that in the control group, *PER2* activity gradually increases over the course of a day with a peak at midnight; no such *PER2* circadian variation was observed in genetically obese mice that had developed hepatic steatosis. In lipid-laden PER2::LUCIFERASE reporter macrophages, nobiletin restored *PER2* amplitude. Consistent with these in vitro properties, RT qPCR data showed that nobiletin significantly upregulated expression of *Clock* and *Dbp* (a circadian clock related transcription factor) in obese mice [84]. The clock-enhancing properties of nobiletin have been postulated to make the polymethoxy flavone a promising candidate for the treatment of postoperative cognitive dysfunction. Doses of 50 mg/kg body weight for 7 consecutive days dramatically down-regulated mRNA levels of Bmal1, *Rev-erbα, Rorα, Rorγ,* and *Per2* and simultaneously attenuated neuroinflammation and cognitive impairment [85].

Intestinal L cells respond to food intake by secretion of the incretin hormone glucagon-like peptide-1 (GLP-1). In rodents, the cells are under circadian regulation, with response being greater at the onset of the dark/feeding period as compared to the light/fasting period. In humans too, responses vary by time-of-day. High-fat diet in rodents causes a loss of the circadian rhythm in the GLP-1 secretory response, an effect that is ameliorated by nobiletin application (0.3% *w/w* with food). In vitro experiments with male colonic murine (m) GLUTag L cell line (a model of the intestinal L cell) confirmed suppression of clock genes by palmitate which was ameliorated by nobiletin (20 µM) treatment [86]. Similarly, primary cultured mouse hepatocytes treated with palmitate were used as a model for fatty liver and insulin resistance. As in the GLUTag L cell line, palmitate treatment resulted in dampened daily oscillations of circadian clock genes *Cry2, Per2, Rev-erbα* and a concomitant disturbance glycolipid metabolism and reduced insulin sensitivity. Treatment with rather high concentration of nobiletin (200 µM) restored clock gene activity to normal levels [87]. The hepatic lipid metabolism imbalance was ameliorated via modulation of the AMPK-Sirt1 signalling pathway, similar effects had been reported previously for quercetin and EGCG [58,61].

Circadian clocks in pancreatic islets derived from type-2 diabetic human donors were shown to display a dampened amplitude and altered synchronization properties. Nobiletin (20 µM) was shown to boost the amplitude of circadian gene expression in cultured pancreatic islets, with a concomitant increase in insulin secretion [88]. Overexpression of *Bmal1* in transgenic β-Bmal1^OV^ mice resulted in enhanced amplitudes of circadian clock in pancreatic islets and increased glucose-stimulated insulin secretion (GSIS). Transgenic mice were protected against obesity-induced glucose intolerance. Administration of nobiletin (10 µM) to Per2^luc/+^ Ins1^GFP/+^ double transgenic isolated islets in vitro, showed that the flavone enhanced the amplitude of *Per2*-driven luciferase oscillations and augmented GSIS. Nobiletin did not augment GSIS in islets from mice that were lacked Bmal1 (β-Bmal1^−/−^ transgenic mice) [89].

## 4. Conclusions and Future Prospects

The importance of the circadian clock in maintaining human health is now widely acknowledged. Dysregulated and dampened clocks may be a common cause of age-related diseases—hearing loss, cataracts and refractive errors, back and neck pain and osteoarthritis, chronic obstructive pulmonary disease, depression and dementia—and metabolic syndrome—the combination of diabetes, hypertension and obesity, which in turn creates an increased risk of getting coronary heart disease, stroke and other conditions that affect the blood vessels. Thus, circadian clocks should be considered as therapeutic targets to mitigate disease symptoms. Clock-enhancing small molecules as novel preventive and therapeutic agents is an area wide open for exploration [90].

Chrononutrition may become a discipline in its own right, e.g., dedicated to the identification of dietary chronobiotics that can prevent or ameliorate chronic diseases and that can help with healthy ageing [91,92]. At present, there is scientific consensus that a diet rich in fruits and vegetables is key to a prevention of a range of noncommunicable diseases that are collectively responsible for 74% of all deaths worldwide. However, there is no agreement on which specific molecule(s) in the diet play the main role. Polyhydroxy flavonoids have been widely considered, but their generally poor bioavailability makes it unlikely that they directly affect human homeostasis—though they may act indirectly, e.g., via the gut-brain axis. The polymethoxy flavone discussed in this review, nobiletin, has shown some encouraging results in initial in vitro screening, and in further in vivo pharmacokinetic and pre-clinical experiments. We speculate that nobiletin might partially mimick the activity of corticosterone. Although many more experiments are needed to fully elucidate its exact mechanism of action, it is a promising candidate with potential as a chronotherapeutic agent.

## Figures and Tables

**Figure 1 molecules-27-07727-f001:**
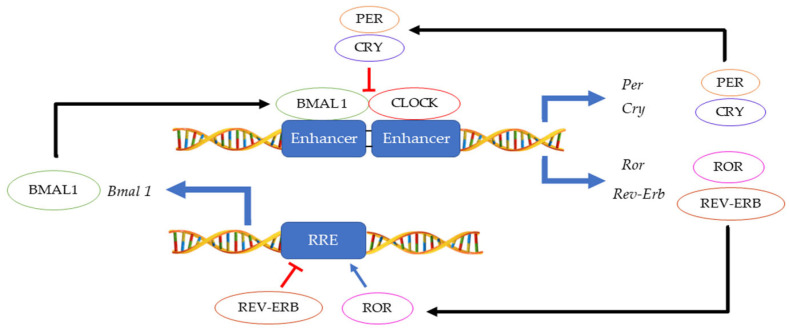
Main feedback loops regulating the circadian clock.

**Figure 2 molecules-27-07727-f002:**
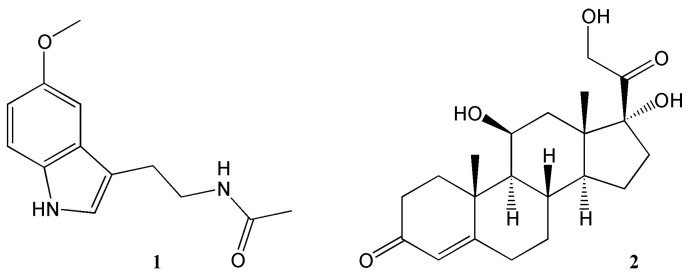
Key hormones regulating circadian rhythms. **1** = melatonin, **2** = corticosterone.

**Figure 3 molecules-27-07727-f003:**
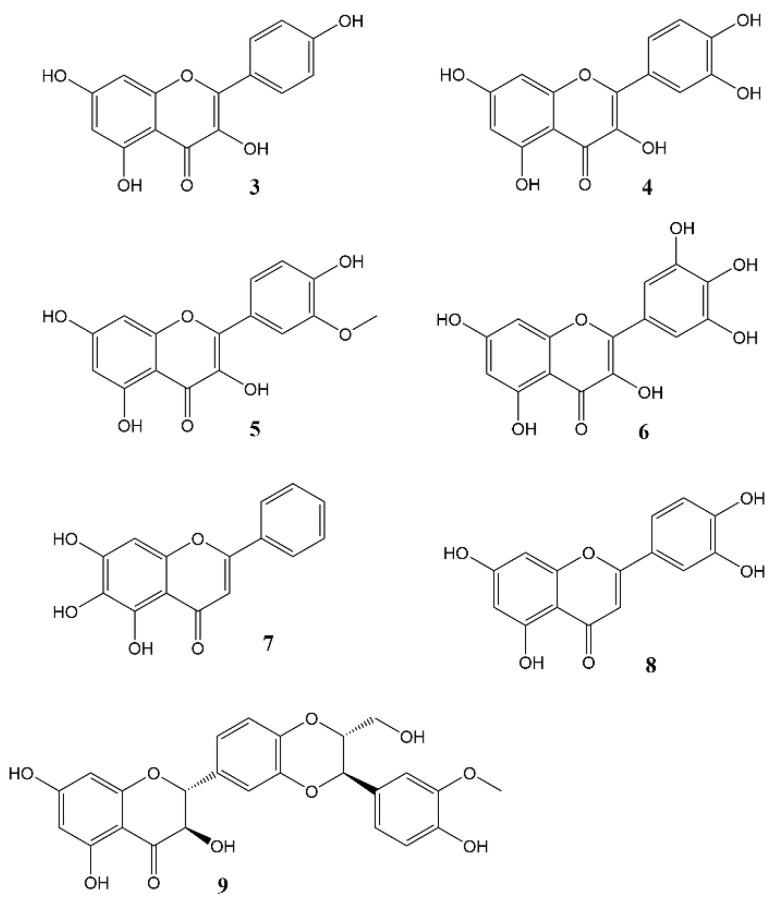
Flavonols and flavones **3** = Kaempferol, **4** = Quercetin, **5** = Isorhamnetin, **6** = Myricetin, **7** = Baicalein, **8** = Luteolin, **9** = Silybin A.

**Figure 4 molecules-27-07727-f004:**
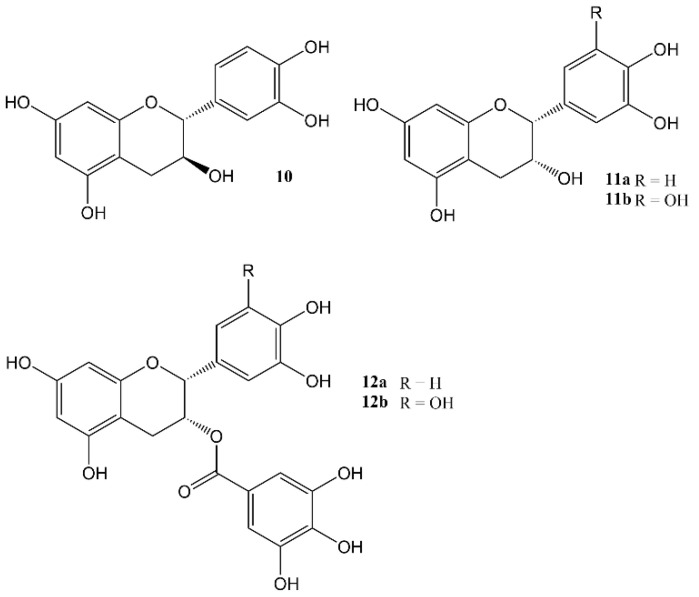
Catechins: **10** = catechin, **11a** = Epicatechin, **11b** = Epigallocatechin, **12a** = Epicatechin gallate, **12b** = Epigallocatechin gallate (EGCG) Proanthocyanidins are di-, tri, tetra-, or oligomers of these units.

**Figure 5 molecules-27-07727-f005:**
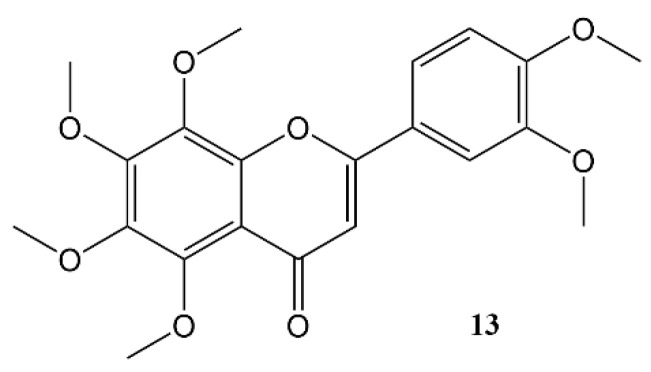
Nobiletin.

## Data Availability

Not applicable.

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
