# Peer review of "Effect of the Citrus Flavone Nobiletin on Circadian Rhythms and Metabolic Syndrome"

_molecules, 2022, doi:10.3390/molecules27227727_

Round 1
Reviewer 1 Report
The title of this article is “Effect of the Citrus flavone nobiletin on circadian rhythms and metabolic syndrome”. This is an interesting topic, and it is an area that needs our attention. However, there are still some areas of the article that need to be revised.
1. Line 139 and 140. The authors mention the regulatory effect of flavonoids on intestinal flora. This part is interesting but not clearly presented by the authors. The authors should explore the relationship between gut flora and circadian rhythm in the context of the close connection between gut flora and brain-gut axis.
2. Please revise the English expressions in your essay by removing unnecessary "the" from the sentences, making sure the sentences look more concise, and replacing words that appear too often in the text.
3. Article "3. Effects of nobiletin circadian rhythm and metabolism". In this part, the authors need to introduce more experimental cases to support the effects of nobiletin.
4. Article 306 and 307 lines. For the section on the role of nobiletin in improving type 2 diabetes, the authors need to go more in depth and give more of their own perspective and outlook for the future.
5. The pictures of the article can be properly optimized, such as adding some relevant organs or adding some colorful pictures. Make the pictures look more vivid.
6. Authors are requested to carefully check the format of the references used in the article to ensure that the references are in the required format.
Author Response
We thank Reviewer 1 for critically reading our manuscript and providing useful and encouraging remarks. We will address the points of critique below;
- Line 139 and 140. The authors mention the regulatory effect of flavonoids on intestinal flora. This part is interesting but not clearly presented by the authors. The authors should explore the relationship between gut flora and circadian rhythm in the context of the close connection between gut flora and brain-gut axis.
We agree that the concept of a brain-gut axis is interesting and worth exploring further. We now mention this prominently in our final conclusion (lines 348-349). Further, we have inserted a section (lines 147-153) wherein we briefly discuss the brain-gut axis concept. We shied away from listing microbiota metabolites that may directly affect circadian rhythms, since this field is still developing and any conclusions would be very preliminary. - Please revise the English expressions in your essay by removing unnecessary "the" from the sentences, making sure the sentences look more concise, and replacing words that appear too often in the text.
We have critically run through the text again and made minor revisions throughout. - Article "3. Effects of nobiletin circadian rhythm and metabolism". In this part, the authors need to introduce more experimental cases to support the effects of nobiletin.
Nobiletin stands out from the other flavonoids because it has been shown to directly affect the circadian system (rather than indirectly via the gut-brain axis), and it has superior pharmacokinetic properties. An additional paragraph is added (lines 239-244) to highlight this. The number of papers cited on the effect of nobiletin (references 75-91) exceeds that of all other flavonoids combined. - Article 306 and 307 lines. For the section on the role of nobiletin in improving type 2 diabetes, the authors need to go more in depth and give more of their own perspective and outlook for the future.
In chapter 3 (lines 242-244) and in the conclusions (lines 351-352) we explore some thoughts from our own perspective. It may be rather speculative, but indicates where we think the role of nobiletin may be. - The pictures of the article can be properly optimized, such as adding some relevant organs or adding some colorful pictures. Make the pictures look more vivid.
We’ve critically looked at out figures and made some minor changes to figures 3-6. Figure 1 was re-drawn in a more vivid and colourful way as suggested. - Authors are requested to carefully check the format of the references used in the article to ensure that the references are in the required format.
We have re-formatted the manuscript according to the author’s instructions by MDPI. The in-text references are now numbered and the reference list is in order of appearance. These format changes are not highlighted. All other changes are indicated in the text using the Word Review option, as requested.
Reviewer 2 Report
This is a comprehensive, well-organized, well-documented and timely review advocating for further research into the chronotherapeutic potential of flavenoid compounds. Overall, it is a rich source of information and literature relevant to the topic, and the general argument relating circadian system function to health, plants to nutritional health and plant-derived flavenoids to circadian system function is potentially important for further understanding circadian systems and their importance for health, including possible pharmacological opportunities and applications. It is likely that this article will stimulate further interdisciplinary research along these lines. I recommend publication with minor revision to make descriptions of circadian mechanisms a bit more precise and to correct a few minor grammatical typos. Also, the title and abstract are specific to nobiletin, but nobiletin is only one of several compounds described and discussed- a more accurate, descriptive, general and useful title would be something like "Chronotherapeutic Potential of Flavenoids". The most important point for revision is to sharpen the precision of the discussion of circadian mechanisms in the introduction to recognize that the SCN-based light-entrainable animal circadian pacemaker is only one of several circadian pacemakers in animals, there are others in other kingdoms, and a strict hierarchical model connecting circadian pacemakers to peripheral oscillators is an obsolete concept, replaced by a circadian system of multiple pacemakers coupled in many ways to numerous peripheral circadian oscillators. A more accurate concept of circadian systems strengthens the author's arguments and the significance of the manuscript because it opens up many more potential mechanisms through which flavenoids may interact with circadian mechanisms.
SPECIFIC COMMENTS
Introduction line 36: change "complex" to complexes?
Introduction, line 41: cyanobacteria etc. are Kingdoms, not species, and the Drosophila model is limited to animals. The other kingdoms have circadian clocks but the mechanisms are analogous, not homologous.
Introduction, line 43 and thereafter: "CLOCK:BMAIL1" is misspelled. It should be "BMAL1". Please correct throughout the manuscript.
Introduction, line 44: Change "transcriptions" to "transcription"?
Introduction, line 48: Change "requested" to 'required"?
Introduction, line 50: delete "the molecules of"?
Introduction, lines 65-66: Change "circadian mechanism" to "circadian system"; change "hierarchical" to "complex"; Change "master oscillator" to light-entrainable pacemaker". There are multiple circadian mechanisms in all organisms. The per:tim-cry/CLOCK:BMAL1 mechanism is a light-entrainable circadian pacemaker specific to animals and is only one of several animal circadian clock mechanisms. Others include food-entrainable, methamphetamine-inducible and Redox circadian pacemakers. The circadian mechanisms in a cell are best described as a "circadian system" with multiple circadian pacemakers and many peripheral circadian oscillators that are coupled among themselves in complex ways that include vertical, hierarchical and mutually entraining pathways. I think the authors are focusing their review on the SCN-based light-entrainable animal circadian pacemaker and should, therefore, be more specific in this respect.
Introduction, line 78: change "entrains the peripheral clock" to "can entrain peripheral clocks"?
Introduction, line 79: change setting cells in a "nocturnal state" to "acting as a signal for the dark phase of the photoperiod".
Introduction, line 106: change "jet lags" to "jet lag" here and throughout the manuscript;
Introduction, line 107: change "jet lags are" to "jet lag is";
Introduction, line 124: change "revenues" to "candidates";
Introduction, one 132: change "stabilising" to "regulating" (e.g. the effects may not always be stabilizing but could be disruptive as well);
Introduction, line 150: change "both animal" to "both the animal?
Introduction, line 151: The fact that flavenoids affect circadian clock mechanisms in both plants and animals does not mean the underlying circadian mechanisms are conserved. The effects may occur in each kingdom through different mechanisms. The analogous mechanism may be a common selection pressure for flavenoids to alter circadian timing- in the case of plants, possibly their own endogenous clocks and rhythms, in the case of animals, possibly the circadian timing of predation or pollination, for example.
Introduction, line 162: change "mice" to "mouse"?
Introduction, line 194: change "an initial in" to "an initial increase in"?
Introduction, line 255: change "done skeletal" to "done in skeletal"?
Author Response
We thank Reviewer 2 for critically reading our manuscript an providing useful commentaries. We will address the points made in the review below:
...the title and abstract are specific to nobiletin, but nobiletin is only one of several compounds described and discussed- a more accurate, descriptive, general and useful title would be something like "Chronotherapeutic Potential of Flavenoids".
In the revised manuscript, we have emphasized more strongly that nobiletin stands out from the other flavonoids because it has been shown to directly affect the circadian system (rather than indirectly via the gut-brain axis), and it has superior pharmacokinetic properties. An additional paragraph is added (lines 239-244) to highlight this. The number of papers cited on the effect of nobiletin (references 75-91) exceeds that of all other flavonoids combined.
Considering the title: we think that reference to nobiletin is more precise and we prefer that over the more general reference to flavonoids – more so since there is doubt whether other flavonoids work via the same mechanism of action.
The most important point for revision is to sharpen the precision of the discussion of circadian mechanisms in the introduction to recognize that the SCN-based light-entrainable animal circadian pacemaker is only one of several circadian pacemakers in animals, there are others in other kingdoms, and a strict hierarchical model connecting circadian pacemakers to peripheral oscillators is an obsolete concept, replaced by a circadian system of multiple pacemakers coupled in many ways to numerous peripheral circadian oscillators.
That is a valid point. We made some alterations (lines 60-65) and added a paragraph (lines 79-85) to do justice to the complexity of the circadian mechanisms. We acknowledge the system of multiple pacemakers coupled in many ways to numerous peripheral circadian oscillators, but rather than go into great detail in our manuscript we refer to two 2020 review papers (References 27, 28).
Introduction line 36: change "complex" to complexes?
Corrected
Introduction, line 41: cyanobacteria etc. are Kingdoms, not species, and the Drosophila model is limited to animals. The other kingdoms have circadian clocks but the mechanisms are analogous, not homologous.
Corrected
Introduction, line 43 and thereafter: "CLOCK:BMAIL1" is misspelled. It should be "BMAL1". Please correct throughout the manuscript.
Corrected throughout the manuscript
Introduction, line 44: Change "transcriptions" to "transcription"?
Corrected
Introduction, line 48: Change "requested" to 'required"?
Corrected
Introduction, line 50: delete "the molecules of"?
Corrected
Introduction, lines 65-66: Change "circadian mechanism" to "circadian system"; change "hierarchical" to "complex"; Change "master oscillator" to light-entrainable pacemaker". There are multiple circadian mechanisms in all organisms. The per:tim-cry/CLOCK:BMAL1 mechanism is a light-entrainable circadian pacemaker specific to animals and is only one of several animal circadian clock mechanisms. Others include food-entrainable, methamphetamine-inducible and Redox circadian pacemakers. The circadian mechanisms in a cell are best described as a "circadian system" with multiple circadian pacemakers and many peripheral circadian oscillators that are coupled among themselves in complex ways that include vertical, hierarchical and mutually entraining pathways. I think the authors are focusing their review on the SCN-based light-entrainable animal circadian pacemaker and should, therefore, be more specific in this respect.
Corrected. Also, see response above to the issue of multiple pacemakers coupled to numerous peripheral circadian oscillators.
Introduction, line 78: change "entrains the peripheral clock" to "can entrain peripheral clocks"?
Corrected.
Introduction, line 79: change setting cells in a "nocturnal state" to "acting as a signal for the dark phase of the photoperiod".
Corrected
Introduction, line 106: change "jet lags" to "jet lag" here and throughout the manuscript;
Corrected
Introduction, line 107: change "jet lags are" to "jet lag is";
Corrected
Introduction, line 124: change "revenues" to "candidates";
Corrected
Introduction, one 132: change "stabilising" to "regulating" (e.g. the effects may not always be stabilizing but could be disruptive as well);
Corrected
Introduction, line 150: change "both animal" to "both the animal?
Corrected
Introduction, line 151: The fact that flavenoids affect circadian clock mechanisms in both plants and animals does not mean the underlying circadian mechanisms are conserved. The effects may occur in each kingdom through different mechanisms. The analogous mechanism may be a common selection pressure for flavenoids to alter circadian timing- in the case of plants, possibly their own endogenous clocks and rhythms, in the case of animals, possibly the circadian timing of predation or pollination, for example.
We left the text as it was. The main point is that an alteration in flavonoid content is concomitant with an alteration in circadian clock mechanisms, irrespective of the mechanism. We don’t know whether this is convergent evolution or not, or whether homologous or analogous changes are involved, but either way flavonoids appear linked with clocks.
Introduction, line 162: change "mice" to "mouse"?
Changed to ‘mouse brains’.
Introduction, line 194: change "an initial in" to "an initial increase in"?
Corrected
Introduction, line 255: change "done skeletal" to "done in skeletal"?
Corrected